# Role of Microalgae in Global CO$_2$ Sequestration: Physiological Mechanism, Recent Development, Challenges, and Future Prospective

Ravindra Prasad [1,*], Sanjay Kumar Gupta [1], Nisha Shabnam [2], Carlos Yure B. Oliveira [3], Arvind Kumar Nema [1], Faiz Ahmad Ansari [4] and Faizal Bux [4]

1 Department of Civil Engineering, Indian Institute of Technology, Delhi 110016, India; sanjuenv@gmail.com (S.K.G.); aknema@gmail.com (A.K.N.)
2 Department of Biophysics, Centre of the Region Haná for Biotechnological and Agricultural Research, Faculty of Science, Palacký University, 78371 Olomouc, Czech Republic; shabnam251@gmail.com
3 Departamento de Pesca e Aquicultura, Universidade Federal Rural de Pernambuco, Recife 52171-900, Brazil; carlos.quiroz@fullbrightmail.org
4 Institute for Water and Wastewater Technologies, Durban University of Technology, Durban 4001, South Africa; faizahmad04@gmail.com (F.A.A.); faizalb@dut.ac.za (F.B.)
* Correspondence: rpkarela@gmail.com

**Abstract:** The rising concentration of global atmospheric carbon dioxide (CO$_2$) has severely affected our planet's homeostasis. Efforts are being made worldwide to curb carbon dioxide emissions, but there is still no strategy or technology available to date that is widely accepted. Two basic strategies are employed for reducing CO$_2$ emissions, *viz.* (i) a decrease in fossil fuel use, and increased use of renewable energy sources; and (ii) carbon sequestration by various biological, chemical, or physical methods. This review has explored microalgae's role in carbon sequestration, the physiological apparatus, with special emphasis on the carbon concentration mechanism (CCM). A CCM is a specialized mechanism of microalgae. In this process, a sub-cellular organelle known as pyrenoid, containing a high concentration of Ribulose-1,5-bisphosphate carboxylase-oxygenase (Rubisco), helps in the fixation of CO$_2$. One type of carbon concentration mechanism in *Chlamydomonas reinhardtii* and the association of pyrenoid tubules with thylakoids membrane is represented through a typical graphical model. Various environmental factors influencing carbon sequestration in microalgae and associated techno-economic challenges are analyzed critically.

**Keywords:** microalgae; pyrenoid; carbon sequestration; carbon emissions; algae

## 1. Introduction

Climate change is a major threat that severely hampers the survival of various plant and animal species as well as humans. The continuous increase in the emissions of several greenhouse gasses (GHGs), including carbon dioxide (CO$_2$), water vapor, methane (CH$_4$), nitrous oxide (N$_2$O), and fluorinated gases, has aggravated climate change [1,2]. The rise in GHGs emissions is mostly associated with anthropogenic actions, with the use of fossil fuels being the largest contributor [3]. The world's atmospheric CO$_2$ has increased from ~313 ppm (in 1960) to ~411 ppm at present [4]. A high level of CO$_2$ in the atmosphere raises the acidity of ocean water and affects the marine ecosystem to a significant extent [5,6]. Hence, it is highly imperative at this moment to develop an appropriate strategy to reduce or stabilize the CO$_2$ content in the atmosphere. Various countries have signed many international protocols to curb GHGs emissions, e.g., COP26, Kyoto Protocol (1997), and the Paris agreement (2015).

Two basic approaches for reducing CO$_2$ emissions include (i) the decreased use of fossil fuels complemented with the increased use of renewable energy sources; (ii) and carbon capture and storage via various biological, chemical, or physical methods [7,8].

The physical methods for carbon emission reduction have been extensively explored. Still, there are several technological and economic limitations with the existing technologies. Therefore, it is crucial to upgrade the existing technologies as well as develop suitable alternatives. Among various others, biological $CO_2$ fixation seems to be a relatively cost-effective and eco-friendly approach in comparison to the physical and chemical methods. Photosynthetic organisms assimilate $CO_2$ via the dark phase of photosynthesis and play a key role in maintaining the balance of $CO_2$ levels in the atmosphere. Compared to other photosynthetic organisms, phytoplankton had higher $CO_2$ fixation efficiency and biomass productivity [6]. Marine phytoplankton accounts for half of the total global primary productivity by fixing ~ 50 gigatons of $CO_2$ annually [6]. In this context, research on $CO_2$ sequestration by microalgae has attracted attention across the globe [9–14]. Microalgae can assimilate $CO_2$ 10–50 times more effectively, compared to vascular plants without competing or providing food to humans/animals [15–17]. Microalgae have a special mechanism to assimilate carbon dioxide known as the carbon concentration mechanism (CCM). In this mechanism, a specialized organelle i.e., pyrenoid increases the concentration of $CO_2$ around the thylakoid membranes [18]. The increased concentration of carbon dioxide around the thylakoid membrane enhances the efficiency of ribulose-1,5-bisphosphate carboxylase/oxygenase (Rubisco), an important photosynthetic enzyme for carbon assimilation or sequestration. Rubisco has a low affinity for carbon dioxide as it has been evolved in high $CO_2$ and low $O_2$ environments, so the pyrenoid constantly provides an environment for enhanced $CO_2$ fixation [18,19]. It is evident from Table 1 that microalgae grown in various cultivation conditions for carbon sequestration showed higher tolerance for increased $CO_2$ concentration. Some of the investigations also reported that microalgae can also be used for flue gas ($NO_x$ and $SO_x$) sequestration [20]. Microalgae have high biomass yield and tolerance for adverse environmental conditions. Therefore, microalgae are considered a potential feedstock for $CO_2$ sequestration and bioenergy production [21,22].

Microalgae are also used for wastewater treatment and biomass production, which can further be exploited for various applications, including biogas, bioplastics, and fertilizer production [23,24]. The low nutrition conditions and high photosynthetic efficiency have made it easy to cultivate algae for their exploitation for various applications. The previously published reviews extensively majorly explored the carbon sequestration of microalgae and their physiological mechanism, however detailed information on the role of pyrenoids in the sequestration of $CO_2$ is missing, which are key aspects with specific reference to the global $CO_2$ mitigation using algal technologies [25,26]. This review comprehensively discusses the physiological mechanism of carbon sequestration, the role of pyrenoids, and the impact of environmental factors in the carbon concentration mechanism. Besides this, the review also provides the current global $CO_2$ emission status and scenarios.

### 1.1. Global $CO_2$ Emission Status

Worldwide progress in the economy and the population boom have led to a continuously rapid rise in the emissions of carbon dioxide in the last few decades [27]. The rising level of $CO_2$ in the atmosphere leads to an increased global average surface temperature, which directly and indirectly influences the global weather and climatic phenomenon (e.g., excessive rainstorms, drought) [27,28]. In order to combat the increasing earth's surface temperature, the Paris agreement came into force, which was ratified by 196 countries to limit global warming below 1.5 °C compared to the pre-industrial era. This can be achieved through reducing greenhouse emissions by Nationally Determined Contributions (NDCs). The global carbon dioxide emissions have increased by 0.9% in 2019 compared to 2018. The largest emitters were China, USA, India, EU27 + UK, Russia, and Japan as per the Emission Database for Global Atmospheric Research (EDGAR) [29,30]. Demographically, these countries comprise 51% of the global population but contribute to ~67% of $CO_2$ emissions. A detailed $CO_2$ emission scenarios of major contributing countries from 1990 to 2020 is published elsewhere [29,30]. Surprisingly, compared to that in 2018, the level of $CO_2$ emission in 2019 increased in China and India but decreased in EU28, the USA,

and Russia (Figure 1)**.** Global carbon emissions showed a 5% drop in the first quarter of 2020 compared to the first quarter of 2019, due to the decline in the demand for coal (8%), oil (4.5%), and natural gases (2.3%). In another report, the daily, weekly, and seasonal dynamics of $CO_2$ emissions were presented and estimated a ~8.8% decrease in the $CO_2$ emissions in the first half of 2020 [29,30]. The decline in global $CO_2$ emissions in 2020 was due to the COVID-19 pandemic, which recorded the most significant decline since the end of World War II [28]. EDGAR estimated that 2020 showed a decline, with global anthropogenic fossil $CO_2$ emissions 5.1% lower than in 2019, at 36.0 Gt $CO_2$, just below the 36.2 Gt $CO_2$ emission level registered in 2013 [29,30].

In 2019, global carbon emissions (fossil fuels) per unit of Gross Domestic Products (GDP) showed a declining trend reaching an average value of 0.298 t$CO_2$/k USD/yrs., while per capita carbon emissions remained stable at 4.93T$CO_2$/capita/yrs., confirming a 15.9% surge from 1990 [29,30] as published by [29,31].

### 1.2. Carbon Sequestration Technologies

There are various physical, chemical, and biological methods in operation for reducing atmospheric $CO_2$ emissions [7,8,32]. The carbon sequestration or fixation strategies are popularly known as carbon capture and storage/utilization (CCS/U). Carbon emission reductions in CCS is carried out in various stages such as $CO_2$ capture, separation, transportation, utilization, and storage. A detailed discussion of all these steps is demonstrated elsewhere [7,8,32]. A major system frequently used for carbon capture comprises (i) pre-combustion, wherein $CO_2$ is removed before combustion and the fuel is broken down to yield synthesis gas, a mixture of $CO_2$ and $H_2$; subsequently, $CO_2$ is separated into various processes, and $H_2$ is used as a clean fuel; (ii) post-combustion, where $CO_2$ is captured after the combustion of fuels using chemical absorption; (iii) oxy-fuel where the fuel is combusted in the presence of pure oxygen to produce high levels of $CO_2$; and (iv) chemical looping combustion, where oxygen carrier (solid metal oxides) particles are continuously circulated to supply oxygen to react with fuel, wherein the combustion of metal oxide and fuel produce metal, $CO_2$, and $H_2O$ [33]. The separation of $CO_2$ from flue gas also plays a vital role in carbon capture and storage technologies. Many separation techniques in operation include absorption/adsorption, membrane separation, and cryogenic distillation [34]. After capture at the source, $CO_2$ needs to be transported to the sink, which requires further various methodologies described elsewhere [35].

## 2. Physiological Mechanism of Carbon Sequestration in Algae

Aquatic photosynthetic organisms, mainly phytoplankton, are responsible for 50% of the global carbon assimilation [36–38]. It has been stated in the literature that 1.0 kg of cultivated microalgae may assimilate 1.83 kg of $CO_2$ [39,40]. There are three different processes i.e., photoautotrophic, heterotrophic, and mixotrophic metabolisms, involved in algae that help $CO_2$ assimilation [41–43]. Microalgae take up inorganic carbon in three different ways: (i) The transformation of bicarbonates into $CO_2$ by extracellular carbonic anhydrase that readily diffuses inside the cells without any hindrance; (ii) straight absorption of $CO_2$ via the plasma membrane; and (iii) direct intake of bicarbonates by resolute carriers in the membrane, also known as dissolved inorganic carbon (DIC) pumps (Figure 1A) [44].

### 2.1. Photoautotrophic Metabolism

The majority of the microalgae are photoautotrophic, requiring inorganic carbon and light to transform (inorganic) $CO_2$ into carbohydrates by photosynthesis. The algae fix $CO_2$ through the Calvin–Benson cycle (Figure 1A), where the enzyme Rubisco plays a key role in converting $CO_2$ into organic compounds [41,45]. In microalgae, the photosynthetic reaction can be classified as a light-dependent reaction and a light-independent or dark reaction (Figure 1B). The first phase of photosynthesis is light-driven, and here light transforms $NADP^+$ and ADP into energy-storing NADPH and ATP molecules [46]. The second

phase, i.e., the dark phase, consists of $CO_2$ fixation and assimilation via the Calvin–Benson cycle in order to create organic compounds (glucose) with the aid of NADPH and ATP, produced in the first phase [47]. Here, Ribulose bisphosphate carboxylase/oxygenase (Rubisco) plays a significant role in the sequestration of $CO_2$ [48,49]. Rubisco catalyzes the conversion of $CO_2$ to 3-phosphoglycerate. However, due to the oxygenase character, Rubisco binds very weakly binds with $CO_2$, which makes it a poor $CO_2$ fixer [48,49]. These phosphoglycerates are then involved in yielding carbohydrates. Furthermore, these phosphoglycerates are mostly used to regenerate RuBP, which is then employed to continue the carbon-fixing cycle. The oxygen ion of Rubisco produces phosphoglycolate, which in turn hinders the carboxylase function of Rubisco. The phosphoglycolate is further transformed into phosphoglycerate (3-PGA) by exploiting ATP and releasing $CO_2$. This reaction is known as photorespiration, in which $O_2$ is utilized and $CO_2$ is released [50]. Therefore, photorespiration leads to the wastage of carbon and energy, eventually decreasing the yield of photosynthesis [51]. Nonetheless, atmospheric $O_2$ concentration usually remains higher compared to atmospheric $CO_2$, thus further favoring the oxygenase functionality of Rubisco and thereby promoting photorespiration. To counter this situation, microalgae have developed $CO_2$ concentrating mechanisms (CCMs) to enhance the concentrations of $CO_2$ within close range of Rubisco [52,53].

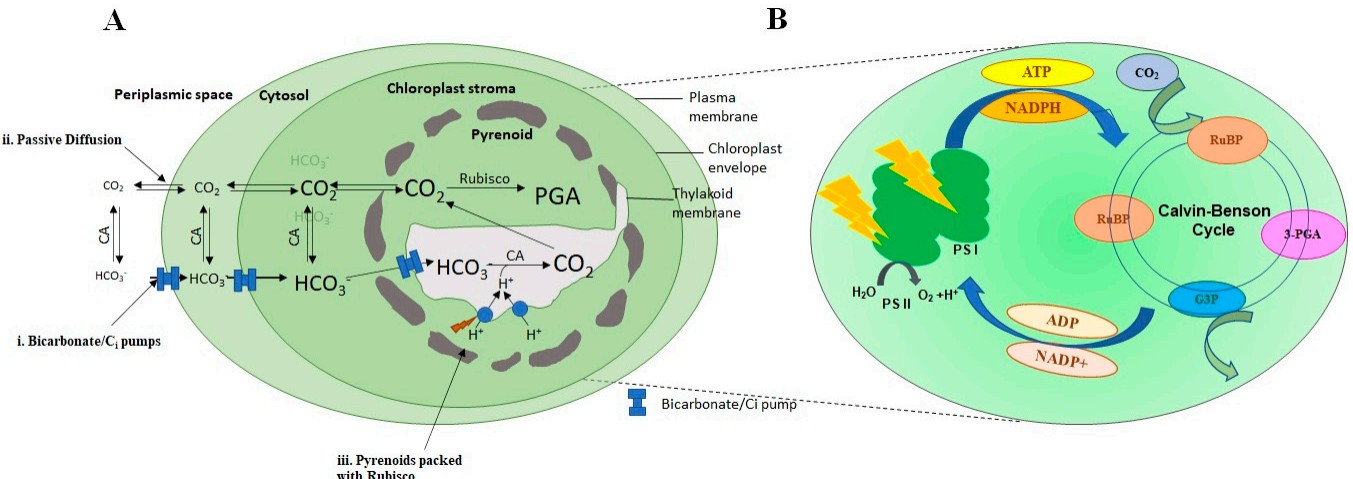

**Figure 1.** Typical figures of carbon concentration mechanism in microalgae *Chlamydomonas reinhardtii* (**A**) showing (i) Bicarbonate/$C_i$ pumps for $C_i$ transportation; (ii) passive diffusion of $CO_2$ through membrane pores; (iii) pyrenoid packed with Rubisco. (**B**) Magnified view of chloroplast where photosynthetic $CO_2$ reduction takes place through Calvin–Benson cycle.

### 2.2. Heterotrophic Metabolism

Heterotrophic metabolism occurs with or without solar energy, and it requires organic carbon. Although the majority of microalgae are photoautotrophic, there are cases where several microalgae can grow via heterotrophic metabolism under dark conditions or under low-light conditions, which is insufficient for autotrophic metabolism. These particular algae heterotrophically metabolize a wide range of organic carbon sources in these light-deprived environments [54–57]. This metabolism follows the pentose phosphate pathway (PPP), which involves the usage of organic carbons derived from acetate, glucose, lactate, and glycerol, and different enzymes involved in transportation, phosphorylation, anabolic and catabolic metabolism, and yielding energy via the substrate or respiration [43]. However, in a few algal strains, heterotrophy can also occur in the presence of light, and such processes are termed photoheterotrophy [58]. The characteristics of the heterotrophic microalgae cultivation are (i) comparably higher capacity to assimilate and grow under light-impoverished conditions; (ii) a fast growth rate; and (iii) the capability to metabolize various types of resources of organic carbon sources [56]. Numerous microalgal strains

have been examined in heterotrophic conditions for the production of biomass, and various important metabolites using glucose as a carbon source [59,60].

### 2.3. Mixotrophic Metabolism

Mixotrophic metabolism obeys both autotrophic photosynthesis and heterotrophic assimilation. This metabolism can be considered a derivative of the heterotrophic metabolism as both $CO_2$ and organic carbon are used together. Mixotrophic metabolism is accompanied by respiration and photosynthesis, resulting in maximum glucose usage [61]. Thus, mixotrophic metabolism can employ both organic and inorganic carbon, thereby leading to the high production of biomass [62,63]. The organic carbon is captured via aerobic respiration, whereas inorganic carbon is absorbed through photosynthesis [64]. Mixotrophic microalgae cultivation delivers higher cell yields per unit of energy input compared to autotrophic or heterotrophic cultivations [65]. Furthermore, mixotrophic metabolism manifests a lower energy-conversion efficiency compared to heterotrophic metabolism [65]. However, both these mechanisms preserve the important pigments and photosynthetic carotenoids under solar irradiation [66,67]. There are certain aspects where mixotrophic cultivation offers extra benefits over photoautotrophic cultivation, such as an increased growth rate, decreased growth cycles, insignificant decrement of biomass in the dark, and overall higher biomass yields [68,69]. However, mixotrophic metabolisms have their own disadvantages, i.e., comparably costly due to the high requirement of organic carbon resources and are vulnerable to intrusive heterotrophic bacteria in bare pond arrangements Moreover, balancing two kinds of metabolisms is another challenge for the mixotrophic mechanism. However, mixotrophic metabolisms have their own disadvantages, as they are costly due to their necessity of organic carbon resources and are vulnerable to intrusive heterotrophic bacteria in bare pond arrangements [70]. Moreover, balancing two kinds of metabolisms is another challenge for the mixotrophic mechanism.

### 3. Carbon Concentration Mechanism (CCM) in Algae

Microalgae have been studied for several decades as important feedstocks for bioenergy production to reduce global carbon emissions [71]. As explained in the previous section, the Calvin–Benson cycle (C3 cycle) is the fundamental photosynthetic carbon metabolic pathway in algae (Figure 1B). $CO_2$ is one of the limiting substrates in the aquatic system [72]. Bicarbonate is the prevailing $CO_2$ form in the water at pH $\geq$ 7 and temperatures < 30 °C . Aquatic carbon capture involves the bicarbonate form, which is further required from the growth of algae and creating biomass. The aquatic photosynthetic organisms are continuously exposed to varying degrees of physicochemical stresses depending upon the water matrix, level of dissolved inorganic carbon ($C_i$, $CO_2$, and/or $HCO_3^-$), and geoenvironmental conditions. Hence, aquatic photosynthetic organisms, including microalgae, have developed carbon concentration mechanisms (CCM) as an adaptive mechanism to maximize the photosynthetic efficiency under low $CO_2$ or inorganic carbon conditions [73]. Various environmental factors, such as temperature, pH, alkalinity, etc., directly influence the rate of inorganic carbon supply to the phytoplankton. At times, the water becomes $CO_2$ deficient due to the slower diffusion rate of $CO_2$ in the water, resulting in comparably lower availability of $HCO_3^-$ in the aquatic environment [74]. The involvement of different CCM strategies in several algal strains has been demonstrated in numerous previous studies [75–77]. Constant research efforts in this field confirm the involvement of different CCM strategies in many algae [75–77]. However, detection of the $CO_2$ deficiency in both intra- and extra-cellular levels in order to understand the structure and biochemistry behind CCMs or to reveal the nature of the signal responsible for CCM activation, like many other unexplored functional aspects of CCMs, is yet to be studied. Furthermore, the CCMs are widely distributed among microalgae, both phylogenetically and geographically, although they are absent in certain microalgal groups, such as chrysophyte and synurophycean. Different organisms have different levels of CCMs. However, there are three main CCMs, *viz.* C4 pathways, inorganic carbon transportation, and the

conversion mechanism, which increase the $CO_2$ concentration around the enzyme and are commonly available in almost all of the organisms to attain the desired level of $CO_2$ concentrations, and they are as follows:

Biophysical CCMs involve the concentration of inorganic C, Rubisco-rich environments. These CCM processes occur before the incorporation of inorganic carbon into organic compounds [78]. Irrespective of being prokaryotic or eukaryotic, algal CCM comprises (i) $C_i$ transporters; (ii) carbonic anhydrase for the conversion of Ci to $CO_2$; and (iii) a microcompartment, packed with Rubisco, where $CO_2$ is delivered. The carboxysome and pyrenoid are the microcompartments where $CO_2$ fixation takes place in prokaryotic and eukaryotic algae, respectively (Figure 1).

### 3.1. C4 Pathways

This CCM is based on the C4 and crassulacean acid metabolism (CAM) pathways, wherein $CO_2$ is assimilated by PEP to generate oxalo acetic acid (OAA), which is later decarboxylated to regenerate $CO_2$. This pathway also recaptures the $CO_2$ generated from the oxygenation of Rubisco via photorespiration, thus maximizing the assimilation of $CO_2$ [52,79]. The C4 pathway is well understood in higher plants. Although the existence of a C4 pathway has been reported in the marine diatom *Thalassiosira weissflogii*, C3 photosynthesis is predominant in algae [80,81].

### 3.2. Inorganic Carbon ($C_i$) Transportation and Conversion Mechanism

Carboxysome, a bacterial microcompartment, serves as the key $CO_2$ fixing machinery in all cyanobacteria and many chemoautotrophs [81–83]. The CCMs in prokaryotes include (i) bicarbonate pumps/transporters and membrane-bound hydration enzymes that concentrate bicarbonate levels in the cytosol; and (ii) carboxysome, a selectively permeable protein shell that encapsulates Rubisco and CA, where enhanced levels of bicarbonate are fed and converted to $CO_2$ to initiate the first step of C3 cycle [75,84]. The encapsulation of Rubisco and CA by the protein shell restricts the leakage of $CO_2$ (converted by CA) from the carboxysome into the cytosol, thus elevating the levels of $CO_2$ around Rubisco [75,83].

### 3.3. Raise of CO$_2$ Concentration around the Enzyme

This is the third type of CCM (Figure 1A) controlled by the pH gradient arrangement over the chloroplast and thylakoid membrane upon illumination. Commonly, Eukaryotic algae follow this sort of CCM. Under light, the chloroplast stroma attains a pH around 8.0, whereas the thylakoid lumen achieves a pH between 4.0 and 5.0 This pH gradient is important as at pKa 6.3, bicarbonate is converted into $CO_2$. However, here, $C_i$'s $HCO_3^-$ form dominates within the chloroplast stroma, albeit the $CO_2$ group of $C_i$ is found in a generous amount in the thylakoid lumen. Moreover, the bicarbonate moved inside the thylakoid lumen would be changed into $CO_2$, thereby raising the $CO_2$ concentration over the normal range. It is reported that this type of CCM needs Carbonic Anhydrases (CA) in the acidic thylakoid lumen in order to transform the infiltrating $HCO_3^-$ into $CO_2$ immediately [85]. Besides, $HCO_3^-$ is unable to rapidly pass through the biological membranes [86]. Hence, a transport protein or complex may be there to help $HCO_3^-$ to penetrate through the thylakoid lumen. Thus, the present model suggests that $CO_2$ would not accumulate in the dark since light-supported photosynthetic electron transport is necessary to create these pH differentials. [37]

In this model, the CCM can be operated by acquiring and delivering the Ci from the environment to the chloroplast stroma and the generation of elevated levels of $HCO_3^-$ in the chloroplast stroma. The CCM model in *Chlamydomonas* can be described in three phases. The first phase includes the acquisition and delivery of Ci from the environment to the stroma through the concerted action of the functioning of CAs in the periplasmic and cytoplasmic space and C transporters on the plasma membrane and chloroplast envelope [87–89]. The second part includes the entry of $HCO_3^-$ to the thylakoid lumen and the generation of high levels of $HCO_3^-$ in the lumen by utilizing the pH gradient

across the thylakoid membrane [87]. Under light conditions, the chloroplast stroma has a pH ~8.0 while the thylakoid lumen has a pH between 4.0 and 5.0, thus establishing a pH gradient across the thylakoid membrane [90,91]. Due to this pH gradient, $HCO_3^-$ is the predominant Ci species in the thylakoid stroma, whereas $CO_2$ is the predominant Ci in the thylakoid lumen. This step is coordinated through CAs located in the stroma and the thylakoid lumen, as well as the Ci transporter on the thylakoid membrane. The third phase is not present in all algae, but is in the majority of algae and takes place exclusively in pyrenoid. Pyrenoid is a membrane-less organelle (or a sub-compartment) found in the stroma of chloroplasts of most but not all algae [92,93]. The pyrenoid consists of (i) a matrix that is densely packed with Rubisco; (ii) a starch sheath that surrounds the matrix; and (iii) membrane tubules that traverse the matrix and are continuous with the thylakoid network [93–98]. The $CO_2$ produced in the thylakoid lumen diffuses to the pyrenoid matrix (via tubules) (Figure 2A–H), where it is fixed by Rubisco, thus minimizing Rubisco oxygenase activity and stimulating the carboxylase activity in the stroma [87,95]. Various modeling studies have revealed that algae with pyrenoids are likely to be more effective in maintaining high $CO_2$ levels around Rubisco than those without pyrenoids [96].

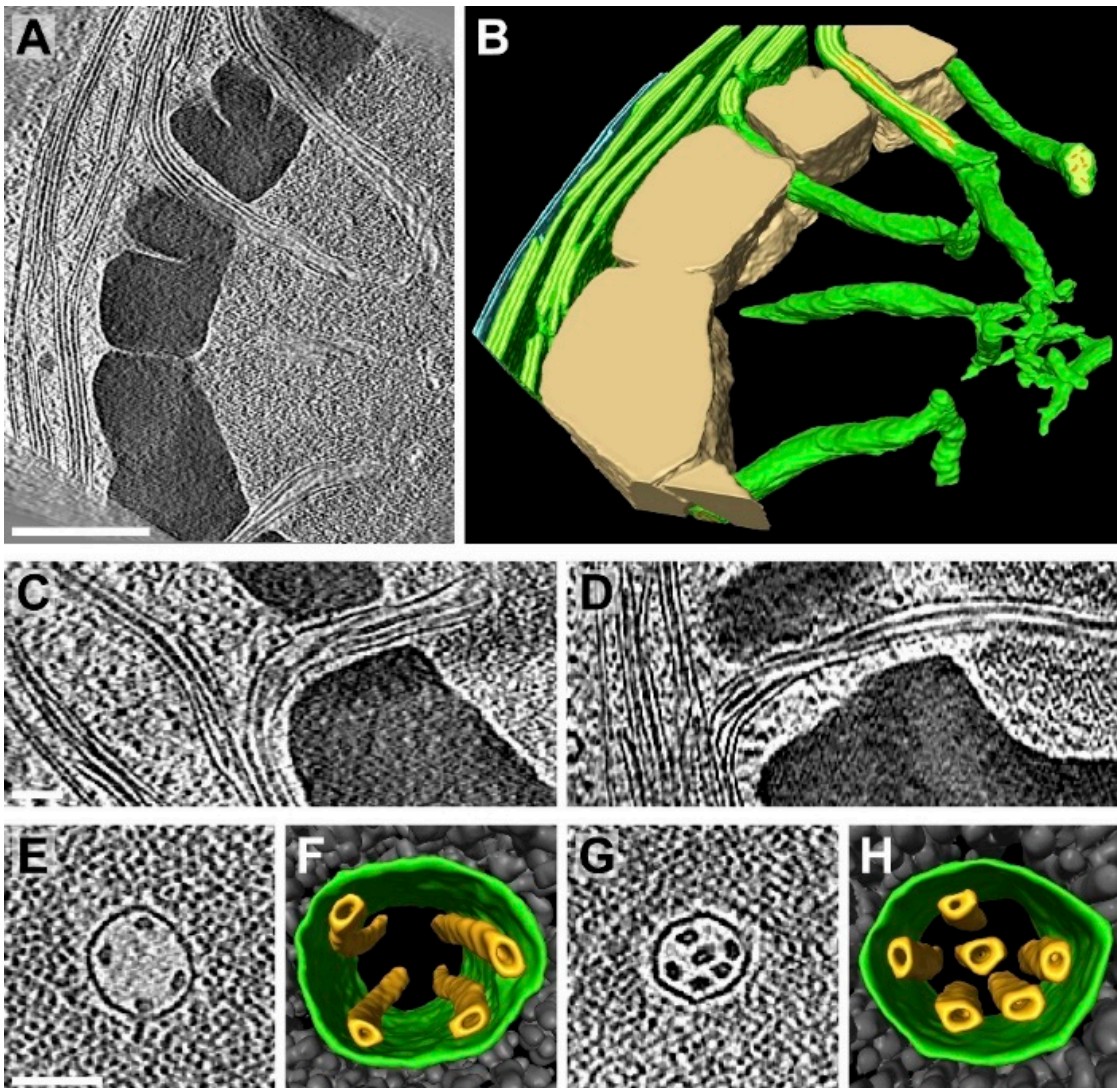

**Figure 2.** A slice of pyrenoids tomographs. (**A**) Tubules of pyrenoid connected with thylakoids membranes. (**B**) 3D representation of (**A**). Dark-green-colored segments are thylakoids membrane while the orange-colored segments are pyrenoids tubules. (**C,D**) The connection of thylakoid stacks with pyrenoids tubules. (**E–H**) High magnification tomographs of Rubisco-complex and their corresponding 3D representation of cylindrical geometry adapted from [98].

## 4. Recent Development in Microalgae Carbon Sequestration

During the 1970s, the U.S. Department of Energy (DOE) initiated research on the application of algal technologies for wastewater treatment using algae, and the biomass produced during the process was used for methane production. Further, the U.S. Department of Energy's Office of Fuels Development (DOE-OFD) funded a program named the 'Aquatic Species Program (ASP), and the major aim of ASP was to evaluate the potential of microalgae in biodiesel production from high lipid-content microalgal biomass grown in ponds, through the utilization of $CO_2$ emitted from coal-fired power plants. Various microalgal species were screened and investigated under different physicochemical conditions i.e., varying temperature, pH, and salinity, etc., but the desired results were not achieved. The two major successes achieved under the aquatic species program included (i) the establishment of a microalgae culture collection canter; and (ii) a pilot-scale microalgae cultivation raceway pond in New Mexico. Fortunately, at the same time, Japanese researchers were also working on a project related to bio fixation of $CO_2$, and greenhouse gas emission abatement using closed microalgae photobioreactors (PBRs), which was later discontinued due to the higher instrumental and maintenance cost of the reactors.

In addition, the US DOE-NETL started PBRs-based microalgae research and development. Moreover, Arizona Public Services, ENEL ProduzioneRicerca, EniTecnologie, ExxonMobil, and Rio Tinto are also taking part the microalgae-based $CO_2$ mitigation research [98,99].

Seambiotic Ltd. was the first to introduce flue gas usage from a power plant to cultivate algae. Seambiotic Ltd., jointly with the Rutenberg coal-fired power plant (based in the city of Ashkelon, Israel), checked the development of the potential of algae exploiting $CO_2$ from flue gas, which is 50% better than the uncontaminated $CO_2$. Numerous other projects are running worldwide, achieving profitable-scale microalgae harvesting amenities by exploiting flue gas. Across the globe, several microalgae-based research projects and programs are in action to develop apt strategies to reduce atmospheric $CO_2$ emissions. The projects are mainly based on reducing the operational cost of carbon sequestration by using waste resources for algae biomass production. Integrated algal biorefinery systems utilize the waste resource *viz.* flue gas and wastewater as a cultivation medium for their growth and development. The biomass produced during the process can be exploited for other valuable derivatives productions [100–158]. However, all the research endeavors so far indicate that utilizing $CO_2$ in the real world by employing microalgae still needs new and creative ideas to secure scientific and technological progress in this field. Table 1 lists the investigations carried out with microalgae for $CO_2$ sequestration under various conditions. It is necessary to merge alternative techniques or co-processes to ensure the expenditure on microalgae research is a remarkable success and resolves the global $CO_2$ problem. Promising alternatives involve wastewater management, the generation of beneficial metabolites, biofuels, animal feed, and biofertilizer creation [103]. The researchers must aim to attain greater biomass productions, cultivation stability, cost-effective cultivation procedures, and advanced biomass-to-fuel transformation methods.

**Table 1.** Chronological order of the studies on $CO_2$ tolerance, sequestration, and sequestration efficiency under various cultivation conditions.

| S NO. | Microalgae | $CO_2$ Tolerance Capacity (%) | $CO_2$ Assimilation Rate (g/L/d) | $CO_2$ Assimilation Efficiency (g/L/d) | Cultivation Conditions | | | | | Cultivation System | Reference |
|---|---|---|---|---|---|---|---|---|---|---|---|
| | | | | | pH | T °C | $CO_2$(%) | Light Intensity | Culture Medium | | |
| 1. | *Chlorella* sp. | 40 | 0.097 | - | 7.5–9 | 30 | 15 | 450 [#] | - | - | [104] |
| 2. | *Chlorella vulgaris* | 18 | - | 76 | 7.2 | 30 | 30 | 1800 [*] | f/2 | APBR[a] | [105] |
| 3. | *Chlorella* sp. | 40 | 2.33 | - | 6.3–9 | 26 | 26 | 100 [$] | Modified freshwater medium | FPBR | [106] |
| 4. | *Chlorella* sp. | 40 | 0.510 | - | 8.2 | 18 | 10–20 | 84 [#] | - | BCPBR | [107] |
| 5. | *Desmodesmus* sp. | 100 | 1.58 | - | - | 30 | 30 | 60 [#] | 3N-BBM | FBC | [108] |
| 6. | *Chlorella vulgaris* | 18 | 2.22 | - | - | 30 | - | 70 [#] | BG11 | BCPB | [109] |
| 7. | *Chlorella pyrenoidosa* *Scendesmus* obliquus | 10 10 | 0.26 0.28 | - - | 7 | 25 | 10 | 180 [#] | BG11 | EF | [110] |

**Table 1.** *Cont.*

| S NO. | Microalgae | CO$_2$ Tolerance Capacity (%) | CO$_2$ Assimilation Rate (g/L/d) | CO$_2$ Assimilation Efficiency (g/L/d) | Cultivation Conditions | | | | | Cultivation System | Reference |
|---|---|---|---|---|---|---|---|---|---|---|---|
| | | | | | pH | T °C | CO$_2$(%) | Light Intensity | Culture Medium | | |
| 8. | *Chlorella* sp. | - | 0.25 1.7 | | 8 | 18 | 0.03 | 6000 * | f/2 and AFW | BPR | [111] |
| 9. | *Anabaena* sp. | 10 | 1.01 | 67–79 | | 20–25 | 5–15 | 127–250 # | BG11 | BPR | [112] |
| 10. | *Scenedesmus obliquus* | 18 | 0.252 | - | 7 | 25 | 13.8 | 5496 * | f/2 | EF | [113] |
| 11. | *Scenedesmus obliquus* | 18 | - | 67 | - | 26 | 26–28 | 12,000 * | Soil extract | APBR | [114] |
| 12. | *Chlorella* sp. | 40 | - | 46 | 10 | 30 | 10 | 30 # | - | LSF | [115] |
| 13. | *Scenedesmus obliquus* | 18 | - | 40.2 | - | 25 | 10 | 12,000 * | - | APBR | [114] |
| 14. | *Botryococcus braunii* | 10 | - | - | - | 25 | 5.5 | 150 # | Chu 13 | - | [116] |
| 15. | *Chlorella vulgaris* | 18 | 0.522 | - | 7.2 | 22 | 22 | 165 # | 3N-BBM | CF | [117] |
| 16. | *Chlorella vulgaris* | 18 | 0.251 | - | 6.0 | 30 | 30 | 3500 * | | FM | [118] |
| 17. | *Chlorella* sp. | 10 | - | - | - | 26 | 10 | 300 # | AFW | BCPBR | [119] |
| 18. | *Chlorella* sp. | 5 | 0.35 | - | 7.18 | - | 5 | 100 # | BG11 | VTPBR | [120] |
| 19. | *Chlorella vulgaris* | 18 | 2.664 | - | 7.02–8.2 | 25 | 25 | 3600 * | Synthetic Sea Salt | PCPB | [121] |
| 20. | *Chlorella vulgaris* | 1 | 6.24 | - | 8.5 | 27 | 0.2 | 75 $ | - | MPBR | [122] |
| 21. | *Chlorella* | 15 | 0.46 | - | 8 | 27 | 0.2 | 200 # | MA | CF | [123] |
| 22. | *Chlorella* | 10 | - | 0.57 | 6 | 25 | 10 | | MBM | BCPBR | [124] |

# in $\mu$mol m$^{-2}$s$^{-1}$, * in lux, $ in $\mu$Em$^{-2}$s$^{-1}$. APBR: Air-lift Photobioreactor, BCPBR: Bubble-Column Photobioreactor, VTPBR: Vertical Tubular Photobioreactor, C.F.: Conical Flask C.F.: Erlenmeyer Flask, LSF: Laboratory Scale Flask, FM: Fermenter, FBC: Feed Batch Cultivation.

## 5. Factor Affecting Carbon Sequestration in Microalgae

### 5.1. CO$_2$ Concentration

CO$_2$ is a primary requirement for microalgae's growth and development to build the carbon skeleton via photosynthesis. Carbon dioxide fixation is carried out by solubilization from the gaseous form to the liquid phase. Carbon dioxide concentration tolerance varies from species to species, as depicted in Table 1. In an investigation, Chlorella vulgaris showed optimal growth at a 5% ($v/v$) CO$_2$ concentration and an inhibitory effect on growth at a 15% ($v/v$) CO$_2$ concentration [125]. Generally, the growth of microalgae at a high CO$_2$ concentration is inhibited due to acidification [126] in the chloroplast stroma region, which inactivates the key enzymes of the Calvin–Benson cycle. A detailed report on the effect of CO$_2$ concentration on individual algal species showed no optimum CO$_2$ concentration to obtain the maximum biomass [127]. In the same investigations, the microalgae showed a high tolerance for carbon dioxide, for example, *Chlorella vulgaris*, Scendesmus obliquus, *Nanochloropsis oculata* Microalgae *Chlorella* sp. T-1, *Scenedesmus* sp., and *Euglena gracilis* studied at different levels of carbon dioxide, i.e., 100%, 80%, and 45%, respectively, showed the high tolerance, but the maximum biomass productivity was found at 10%, 10–20%, and 5%, respectively [127]. Similarly, *Chlorella* sp. KR-1 showed the highest growth rate at 10% CO$_2$ but could tolerate up to 70% CO$_2$. It is worth noting that the reported values represent the solubility of a single component in water without the others' presence. Data for the solubility of individual components of flue gas in the entire flue gas presence are not available. Moreover, as the dissolved gases reacts with water and oxygen, new species are formed that have their own solubility and potential impact on the algal growth system. Therefore, bench-scale testing is necessary to determine the solubilities of the combined flue gas constituents. The range mentioned above is important because certain microalgae can grow with 10–15% carbon dioxide [42]. The range is normally found in flue gases, an important greenhouse gas source of pollution, but may be an excellent carbon dioxide source for microalgae [158].

### 5.2. pH

The microalgae cultivation medium's pH plays a significant role in regulating nutrient uptake, photosynthetic activity, and carbon sequestration efficiency. Microalgal culture mostly shows optimum growth in the ranges of pH 6.5–8.5 except for certain cyanobacteria. However, alkaline (High) pH is beneficial for CO$_2$ absorption because alkaline pH increases the availability of free-CO$_2$ in the culture medium, which favors the growth of

high-$CO_2$-tolerant algae [128]. A recent study reported that at pH 10.6–7.0 and a carbon concentration below 9.52 mmol/L, pH does not have a significant effect on carbon absorption; above the same concentration, pH has a significant effect on carbon absorption [129]. In general, bicarbonate ($HCO_3^-$) is used as a carbon source in the inorganic cultivation medium. $HCO_3^-$ is converted into $OH^-$ by carbonic anhydrase, thus increasing the pH and subsequently altering the equilibrium of different inorganic carbon ($C_i$) species. It has been well documented that flue gas containing a high concentration of $CO_2$ (10–20%) acidifies the algae cultivation medium, thus suppressing the algal growth [90].

*5.3. Temperature*

Temperature is an important factor in the growth and development of a photosynthetic organism. Microalgae have been reported to grow under temperatures ranging 5–40 °C, with the optimum temperature ranging 0–30 °C [130]. The optimum temperature varies from species to species, and the optimum temperature for *Nannochloropsis* oculata is 20 °C and *Chlorella vulgaris* is 30 °C [131]. A low temperature inhibits the carbon sequestration process in algae by reducing the enzymatic activity of ribulose-1,5-bisphosphate (Rubisco) and carboxylase; the enzyme plays an important role in photosynthesis and photorespiration. Similarly, extremely high temperatures alter the metabolic activity and photorespiration of microalgae [132]. In general, the adverse effect of temperature on microalgae is seen above 40 °C, which includes charge separation of PSII and inactivation of oxygen-evolving capacity. The inactivation of oxygen evolution activity occurs due to the conformational change in the 33-kDa protein [133]. The temperature has an inverse relation with $CO_2$ dissolution in the liquid medium, whereby an increase in temperature decreases the $CO_2$ dissolution in the algae cultivation medium.

*5.4. Irradiance*

Light is a mandatory requirement for the bioconversion of $CO_2$ by photoautotrophic and mixotrophic microalgae. There are two phases of the photosynthesis reaction light-dependent reaction and light-independent reaction. The light-dependent reaction converts the photon molecules into the biochemical compound in ATP and NADPH. It is used for carbon sequestration or fixation in the presence of Rubisco in the Calvin–Benson cycle [134]. Therefore, optimum irradiance is required to carry out carbon sequestration (Calvin–Benson cycle) in microalgae [56]. The low and high light intensities lead to alteration in the photosynthetic rate. When there is low light energy, microalgae tend to increase the light capture apparatus (chlorophylls) as efficiently as possible. On the other hand, high light intensities alter the acceptor-side activity of PS II and block the flow of electron transfer from $QA^-$ to $QB^-$; in this process, increased charge recombination promotes the formation of P680, which interacts with the oxygen and forms the singlet oxygen [135]. Two well-known phenomena, photoacclimation and photo-limitation, are involved in the use of light energy by microalgae (i) [136,137]. The photoacclimation concept was defined as a gradual reduction of the photosynthetic pigments (mainly chlorophylls a and b) in response to increased irradiance. In the case that an increase in the pigments in cells occurs, it is possible the photon-molecules are not accessible for all cells, and they need to expand the photosynthetic apparatus [138]. Due to its highly reactive nature, singlet oxygen caused oxidative damage in the PS II. It is well established that singlet oxygen or reactive oxygen species (ROS)-induced oxidative stress is responsible for the low photosynthetic rate and, consequently, low carbon fixation and biomass yield under high light stress [139,140]. Similarly, the relationship between irradiance and photosynthesis was investigated in a dynamic model of photosynthesis, the so-called 'flashing light effect'. In addition to the irradiance, the light period (photoperiod) also affects the growth and biomass content in microalgae. The specific growth rate of *Botryococcus braunii* and *Tetradesmus* (*Scenedesmus*) *obliquus* were found to increase under continuous light irradiance while the growth of *Neochloris texensis* was stimulated under a light/dark cycle under the same irradiance in a

published detailed review on the enhancement of light intensity leading to high biomass productivity [141,142].

## 6. Techno-Economic Challenges with Microalgae

It has been well documented that microalgae can curb $CO_2$ emissions (Table 1), but the carbon sequestration depends on various other factors discussed in the separate sections. The challenges associated with algae-based carbon sequestration include the cultivation system, microalgal strain, flue gas composition, $CO_2$ tolerance capacity, etc. [144]. The operating cost and energy consumption are higher in the biomass cultivation stage [143]. Various studies showed that an open pond is more cost-effective for algae cultivation, but it is not a good choice for maintaining the purity of cultures. The heat transfer, irradiance, and nutrient availability dynamics have been studied and revealed that thermal modeling is essential for the open raceway as detailed in [145]. The flat-plate photobioreactor provides high biomass productivity and consumes low energy [146]. Life cycle analysis (LCA) is a good measure of input–output inventory, energy consumption, cost, and the complete life cycle. The LCA of microalgae carbon sequestration begins with microalgae cultivation and leads to the formation of the product. A comprehensive LCA on microalgae production and flue gas sequestration under outdoor cultivation in raceways ponds showed that the semicontinuous cultivation system displayed a 3.5-times higher growth rate for biomass productivity as compared with batch cultivation. The study also reported that flue-gas-fed outdoor raceways ponds could reduce 45–50% of GHGs emissions than the base case [147]. Another LCA report showed that the biomass production cost is 4.87 US\$/kg and the energy consumption is 0.96 kWh/kg of *Chlorella* biomass. A study reported that 4000 m$^3$ algae cultivation ponds could sequester up to 2.2 k tones of $CO_2$ per year under natural daylight conditions [148]. Another study reported that 50 MW power plants could generate ~414,000 t/yrCO$_2$, and a 1000-ha open raceway pond could mitigate ~250,000 t/yrCO$_2$. In this particular study, algae could reduce 50% of $CO_2$ [149]. However, axenic cultures are preferred in the food and pharmaceutical industries. One of the major challenges in the direct capture of $CO_2$ from the industrial flue gas using microalgae is that flue gas contains 142 chemical compounds that might be toxic for the microalgae [148,150]. The low concentration of NO$_x$ and SO$_x$ in the flue gas can serve as a source of nutrients for microalgae, but higher concentrations can cause toxicity. The solubilization of $CO_2$ in the culture medium is mainly dependent on pH, temperature, and salt concentration. At high temperatures, the solubilization of $CO_2$ in the medium decreases. Therefore, it is important to maintain a cool culture medium. Many culture media contain high salt concentrations, which increase the osmotic pressure and consequently reduce the solubilization of $CO_2$ in the medium. Strategies such as fed-batch cultures can be adopted to gradually provide the salts necessary for microalgal growth [151]. The specialized suitability of algal $CO_2$ sequestration has been shown in various studies; in any case, the significant difficulties are the key and all-encompassing turn of events of advancements that will improve the monetary practicality of algal $CO_2$ sequestration and make this a reasonable modern way to deal with GHG remediation.

## 7. Future Prospective

Microalgae carbon sequestration is a sustainable approach for global $CO_2$ emission reduction [152]. The recent development in cultivation techniques, harvesting, $CO_2$ sequestration capacity, and LCA studies make microalgae a more suitable candidate for carbon emission mitigation [152]. Most studies have focused on finding ways to select and culture various promising microalgae species for efficient $CO_2$ sequestration. Little attention has been given to the development of large-scale and commercial-scale exploitation of carbon sequestration using microalgae. Nonetheless, it is essential to consider and exploit the conditions that influence the microalgae's $CO_2$ capture capacity when it is scaled up for commercialization. Indeed, more extensive research on the industrialization of microalgae science in the practical world is the necessity of this moment. Though the research in

algae carbon sequestration mainly involves modification in the strain, improving cultivation techniques, and harvesting, there is a gap in the area involving factors influencing the carbon sequestration (e.g., change in the light intensity during the day (month, and year), etc.). Similarly, pyrenoid is the most important sub-cellular organ of microalgae that plays a significant role (as described above in a separate section) in the carbon concentration mechanism, but to date, the research in this area is fragmentary. Besides, it is necessary to investigate the high-$CO_2$-tolerant microalgae to enhance carbon sequestration. Further studies must be directed to resolve the minimization techniques of the loss of remaining untouched carbon [153]. Life cycle estimation designs should be built to calculate the atmospheric repercussions of microalgae-based carbon sequestration [154,155]. Nonetheless, until now, most research is based on lab-based research under restricted conditions (Table 1), hence the inadequate amount of information accessible on the vying interplay with other microalgal species and reactor scale-up. There is abundant research devoted to closed systems. However, more studies need to be carried out on open pond systems as these are the exceedingly cost-effective alternative for extensive arrangement. To boost microalgal metabolism, an increased amount of $CO_2$ delivery is a necessary but costly part of microalgae harvesting. Thus, additional research is desirable to find possible low-cost choices [42]. Bio-geo-engineering may enhance the capacity of algae in reducing the escalation in oceanic $CO_2$ and temperature as well. Apart from this, the function of algae is to increase the current sequestration of $CO_2$ as organic carbon over hundreds and hundreds of years at the bottom of the sea and in aquatic sediments, diminishing the rate of augmentation of atmospheric $CO_2$ and at least lessening 'ocean acidification' and the radiative temperature rise by the greenhouse effect of atmospheric $CO_2$. Hence, more attention is required to understand the physiological mechanism of microalgal carbon sequestration with special emphasis on the carbon concentration mechanism [156,157].

## 8. Conclusions

During the investigation, it was realized that carbon sequestration studies using microalgae, mainly tested at a small scale in a laboratory-controlled environment, showed the potential for carbon assimilation. Moreover, a few studies carried out at a large and commercial scale appear promising in reducing major greenhouse gas $CO_2$ emissions and mitigating global warming. It has been determined throughout the study that microalgae have a higher $CO_2$ tolerance, carbon assimilation efficiency, photosynthetic efficiency, and growth rate than terrestrial plants. Carbon sequestration studies of microalgae also focused on improving CCM, Rubisco, pyrenoid, and photosynthetic machinery. The review systematically analyzed factors influencing the growth of microalgae and found that microalgae have a wide range of tolerance and sensitivity for temperature, pH, irradiance, and nutritional condition, but small changes in the cultivation condition alter the product yield. The cultivation technologies of microalgae also play an important role in large-scale production, and recent advancement in the biorefinery-based cultivation system makes us more hopeful for large-scale carbon sequestration. Lastly, the sustainability and economic feasibility of microalgae carbon sequestration at a large scale depend on understanding the photosynthetic mechanism, improving growth factors, and technical infrastructure.

**Author Contributions:** Conceptualization, funding acquisition, writing—original draft preparation, R.P.; writing, editing, revision and analysis, S.K.G.; writing, and review, N.S., C.Y.B.O., & F.A.A.; Supervision, and conceptualization, A.K.N.; Review and editing, F.B. All authors have read and agreed to the published version of the manuscript.

**Funding:** This article is financially supported by grant no. 09/086 (1365)/2019/EMR-I Council of Scientific and Industrial Research (CSIR) under the research associate scheme.

**Institutional Review Board Statement:** Not applicable.

**Informed Consent Statement:** Not applicable.

**Data Availability Statement:** All the data is provided in this manuscript.

**Acknowledgments:** R.P. is thankful to the Indian Institute of Technology (IIT) Delhi, India for providing space to carry out this study. N.S. is thankful for the fellowship under the project "Supporting the International Mobility of Researchers - MSCA-IF at Palacky University Olomouc II (CZ.02.2.69/0.0/0.0/19_074/0016220)".

**Conflicts of Interest:** The authors declare no conflict of interest, and no written or informed consent is required for this article.

## Abbreviations

| | |
|---|---|
| CCM | Carbon Concentration Mechanism |
| CCS | Carbon Capture and Storage |
| Sp. | Species |
| $CO_2$ | Carbon Dioxide |
| NADPH | Nicotinamide Adenine Dinucleotide Phosphate Hydrogen |
| ATP | Adenosine triphosphate |
| CA | Carbonic Anhydrases |

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
