# Peer review of "Role of Microalgae in Global CO2 Sequestration: Physiological Mechanism, Recent Development, Challenges, and Future Prospective"

_sustainability, doi:10.3390/su132313061_

Round 1
Reviewer 1 Report
What is the novelty and originality of this work? Which should be clarified in the introduction
None reference from the Sustainability journal was added, therefore it does not present relevance with this journal
What is the reason why figure 1 does not contain updated data for the year 2020?
At several places, typographical errors are noted, check the subscripts through the document, e.g. CO2
Therefore, I cannot recommend the submitted manuscript is published in Sustainability in this way.
Author Response
Reviewer 1
Comments and Suggestions for Authors
Comment 1. What is the novelty and originality of this work? Which should be clarified in the introduction
R.1 The main focus of this review is to provide updated information about the role of microalgae in carbon sequestration and, more importantly, the physiological mechanism. In addition to this, we have also highlighted the role of pyrenoid in carbon sequestration and factors that affects the carbon concentration mechanism. Most of the recent literature majorly focused on the carbon sequestration strategies and lack of information about the role of pyrenoids, physiological mechanism, etc.
Comment 2. None reference from the Sustainability journal was added, therefore it does not present relevance with this journal
Response: We have collected and included literature using specific terms, e.g., carbon sequestration, biological carbon sequestration irrespective of the journal name or publisher.
Comment 3. What is the reason why figure 1 does not contain updated data for the year 2020?
Response: As per the suggestions, the latest information is included in the revised manuscript.
Comment 4. At several places, typographical errors are noted, check the subscripts through the document, e.g., CO2
Response: All the typographical and grammatical errors are checked and corrected.
Reviewer 2 Report
After going through the manuscript "Role of microalgae in global CO2 sequestration: physiological mechanism, recent development, challenges and future prospective", I would give my comments below.
The manuscript needs to be organized; to improve the quality, the following recommendations can be incorporated.
What makes this review different from the other and from the most recent ones?
Some parts of the manuscript are not standard. for example, the Introduction is very poor. The introduction needs to be more developed, and investigate more than 20-30 papers.
Should be provided comprehensive tables between all of the microalgae in global CO2 sequestration till now used. Along with presenting the advantages and disadvantages and findings.
Section of drawbacks could be increased quality of the manuscript.
A review paper not only should summarize recently published works, but also should contain critical and comprehensive discussions. Therefore, check writing for the whole manuscript. The review should not be presented by listing what has been done by others.
There are some grammatical errors, please carefully check the whole manuscript. the manuscript needs to check again by native.
The article needs a major rewrite. The article is not acceptable in this form.
Author Response
Reviewer 2
After going through the manuscript "Role of microalgae in global CO2 sequestration: physiological mechanism, recent development, challenges, and future perspective", I would give my comments below.
The manuscript needs to be organized; to improve the quality, the following recommendations can be incorporated.
Comment 1. What makes this review different from the other and from the most recent ones?
Response: The review is quite different from the recent literature. It provides updated information about the role of microalgae in carbon sequestration and physiological mechanism. We have also incorporated the information related to the role of pyrenoids in carbon sequestration. Most of the recently published literature does not have much information about the role of pyrenoids in carbon concentration mechanisms.
Comment 2. Some parts of the manuscript are not standard. for example, the Introduction is very poor. The introduction needs to be more developed and investigate more than 20-30 papers.
Response: Thanks for your valuable suggestions. We have revised the whole manuscript.
Comment 3. Should be provided comprehensive tables between all of the microalgae in global CO2 sequestration till now used. Along with presenting the advantages and disadvantages and findings. Section of drawbacks could be increased quality of the manuscript.
Response: The elaborated information in form of a table with references is provided in the table 1.
Comment 4. A review paper not only should summarize recently published works, but also should contain critical and comprehensive discussions. Therefore, check writing for the whole manuscript. The review should not be presented by listing what has been done by others.
Response: We have critically and thoroughly analyzed and discussed the information based on the available literature in the revised manuscript.
Reviewer 3 Report
The manuscript presented the review about the use the microalgae in global CO2 sequestration. This problem is very important and will be of interest to readers. The authors analyzed the modern state and future perspectives or research topics, using modern literature. The manuscript is well written and very clear.
Remarks to the authors.
I recommend it for publication with minor revision.
Major comments:
I have a problem during the revision because the authors did not use the template properly.
Minor comments:
Abstract: …the physiological mechanism, with special emphasis on the carbon concentration mechanism – use the synonym
Pages 2,4, lines 169-170: Physical methods, Greater emphasis, Sources, Greenshouse Gas – Why are you using capital letters?
CO2in – correct to CO2 in
Pages 3-4
[Fig.1] and (Fig. 2). Please, correct
Page 8: Figure 2 - correct Typical figures of to Typical figures of CO2 utilization in algae cell for example. It is necessary to write the designation in Figure 2 according to the legend (a, b, i, and so on).
Page 9: Figure 3 – make all figures designation in one style. In Figure 2 you used small letters, in Figure 3- capital. Delete two dots at the end.
Line 76: delete (i)
Author Response
Reviewer 3
The manuscript presented the review about the use the microalgae in global CO2 sequestration. This problem is very important and will be of interest to readers. The authors analyzed the modern state and future perspectives or research topics, using modern literature. The manuscript is well written and very clear.
I recommend it for publication with minor revision.
Comment 1. I have a problem during the revision because the authors did not use the template properly.
Response: Thanks for the suggestion, the appropriate template is used in the revised manuscript.
Comment 2. Abstract: …the physiological mechanism, with special emphasis on the carbon concentration mechanism – use the synonym
Response: Corrected as per the suggestions.
Comment 3. Pages 2, 4, lines 169- 170: Physical methods, Greater emphasis, Sources, Greenhouse Gas – Why are you using capital letters?
Response: Corrected as per the suggestions.
Comment 4. CO2in – correct to CO2 in Pages 3-4
Response: Corrected as per the suggestions.
Round 2
Reviewer 1 Report
the authors respond satisfactorily to all questions
Reviewer 2 Report
The comments of my first report have been addressed by the authors.